# Risk-guided maternity care to enhance maternal empowerment postpartum: A cluster randomized controlled trial

Jacqueline Lagendijk[1]*, Meertien K. Sijpkens[1], Hiske E. Ernst-Smelt[1], Sarah B. Verbiest[2], Jasper V. Been[1,3,4], Eric A. P. Steegers[1]

**1** Department of Obstetrics and Gynaecology, Erasmus MC, University Medical Centre Rotterdam, Rotterdam, the Netherlands, **2** UNC School of Social Work, University of North Carolina at Chapel Hill, Chapel Hill, NC, United States of America, **3** Division of Neonatology, Department of Paediatrics, Erasmus MC–Sophia Children's Hospital, University Medical Centre Rotterdam, Rotterdam, the Netherlands, **4** Department of Public Health, Erasmus MC, University Medical Centre Rotterdam, Rotterdam, the Netherlands

* j.lagendijk.2@erasmusmc.nl

**Data Availability Statement:** All anonymized data files are available from the 'DANS KNAW database' (https://dans.knaw.nl): https://doi.org/10.17026/dans-xx7-drrx.

## Abstract

### Objective

To investigate whether a structured inquiry during pregnancy of medical factors and social factors associated with low socioeconomic status, and subsequent patient-centred maternity care could increase maternal empowerment.

### Design

Cluster-randomised controlled trial.

### Setting

This study was conducted among pregnant women in selected urban areas in the Netherlands. This study was part of the nationwide Healthy Pregnancy 4 All-2 programme.

### Population

Pregnant women listed at one of the sixteen participating maternity care organisations between July 1, 2015, and Dec 31, 2016.

### Methods

All practices were instructed to provide a systematic risk assessment during pregnancy. Practices were randomly allocated to continue usual care (seven practices), or to provide a patient-centred, risk-guided approach to addressing any risks (nine practices) identified via the risk assessment during pregnancy.

### Main outcome measures

Low postpartum maternal empowerment score.

**Funding:** This study is funded by the Ministry of Health, Welfare and Sports (grant number: 323911). JVB is funded by personal fellowships from the Erasmus MC and the Netherlands Lung Foundation (4.2.14.063JO). The funding sources had no role in the design of this study or its execution, analyses, interpretation of the data, or decision to submit results.

**Competing interests:** The authors have declared that no competing interests exist.

**Abbreviations:** 95% CI, 95% Confidence Interval; aOR, adjusted odds ratio; C-RCT, Cluster-randomised controlled trial; EPDS, Edinburgh Postnatal Depression Scale; ICC, Intraclass correlation coefficient; ITT, Intention to treat; HP4All, Healthy Pregnancy 4 All; MCA, Maternity care assistant; MEQ, Maternal empowerment questionnaire; MIDI, Measurement Instrument for Determinants of Innovations; M2C, Mind2Care; OR, Odds ratio; SE, Standard Error; SES, Socioeconomic status; VIF, Variance inflation factor; WHO, World Health Organisation.

## Results

We recruited 1579 participants; 879 participants in the intervention arm, and 700 participants in the control arm. The prevalence of one or more risk factors during pregnancy was similar between the two arms: 40% and 39%, respectively. In our intention-to-treat analysis, the intervention resulted in a significant reduction in the odds of having a low empowerment score [i.e. the primary outcome; adjusted OR 0.69 ((95% CI 0.47; 0.99), P 0.046)].

## Conclusions

Implementation of additional risk assessment addressing both medical and social factors and subsequent tailored preventive strategies into maternity care reduced the incidence of low maternal empowerment during the postpartum period. Introducing this approach in routine maternity care may help reduce early adversity during the postpartum period.

## Introduction

Maternity care refers to the safe and high quality health care given in relation to pregnancy and delivery of an infant. The purpose of maternity care is manifold: providing information and emotional support, providing adequate care for mother and child, and enhancing maternal empowerment [1, 2]. Empowerment is defined as achieving self-efficacy, and it reflects the process through which people gain greater control over decisions and actions affecting their health [3]. Maternal self-efficacy is a belief held by mothers of their capabilities to perform specific tasks in taking care of the newborn. This self-efficacy, or empowerment, is an important predictor of a successful transition to motherhood [4]. Reduced self-efficacy postpartum, or a low maternal empowerment score, reduces the practice of exclusive breastfeeding, the quality of care provided to the newborn, and it increases the risk of maternal depression [4–6].

Women with a lower socioeconomic status (SES) tend to have lower self-efficacy than women with a higher SES [7–10]. This lower self-efficacy of women with a low SES may further augment their already increased risk of developing adverse maternal and child health outcomes [11–16]. As such, building women's sense of autonomy and control may help reduce inequalities in maternal and child health outcomes resulting from factors related to a person's SES [13, 15, 17–21].

Unfortunately, missed opportunities occur in daily practice as SES indicators and factors related to a person's self-efficacy are insufficiently acknowledged or taken into account in the provision of maternity care [22]. Moreover, there is a consistent inequity in postpartum care provision in the Netherlands, where more care is provided to women with a higher SES, who need it less, than to women who are disadvantaged [23].

We hypothesised that a structured inquiry of medical factors and social factors associated with low SES, and subsequent patient-centred maternity care tailored to these factors during pregnancy and the postpartum period, would increase postpartum maternal empowerment.

## Materials and methods

We conducted a cluster-randomised controlled trial (C-RCT) in six municipalities in the Netherlands to assess the effectiveness of a complex intervention to advance maternal empowerment in the postpartum period [24, 25]. The intervention consisted of a systematic risk

assessment including both medical and social factors, and a patient-centred, risk-guided approach to address risks identified during pregnancy and the postpartum period.

This C-RCT was embedded in the national Healthy Pregnancy 4 All-2 (HP4All-2) programme [24], a successor to the HP4All-1 programme [26]. Both programmes aimed to improve perinatal health by reducing health inequalities during pregnancy and childbirth. Poor neighbourhoods in the Netherlands were the main location to reach women with a lower SES [24]. We used the CONSORT statement for C-RCTs to guide reporting of our findings, and included the CONSORT-flowchart for C-RCTs (Fig 1) [27].

## Study setting

The authors confirm that all ongoing and related trials for this intervention are registered. We registered our study with the Netherlands Trial Registry (NTR 6311) in March 2017, this was after patient enrollment had started (July 2015) but before the end of the inclusion period (August 2017). The study proposal had been reviewed by the METC Erasmus MC in March 2015. As a result of this review, the Committee informed us that the rules laid down in the Medical Research Involving Human Subjects Act (also known by its Dutch abbreviation WMO), did not apply to this research proposal.

## Ethical considerations

We have obtained multi-site approval from the Daily Board of the Medical Ethics Committee Erasmus MC (METC 2015–156). Ethical approval to collaborate with the board of each municipality and with the board of maternity care organisations was provided. In addition, we obtained written informed consent from all three levels of participants. Written informed consent was obtained from every participant prior to inclusion. In addition, we obtained written informed consent from the board of every participating municipality ("Rotterdam", "Schiedam", "Groningen", "Utrecht", "Almere", and "Arnhem"), and from the board of every participating maternity care organisation ("Kraamzorg Rotterdam", "Careyn Kraamzorg", "Kraamzorg Sara", "Kraamzorg de Eilanden", "Kraamzorg de Bakermand", "Kraamzorg het Groene Kruis", "de Kraamvogel", "Zin Kraamzorg", "Yunio", "KraamInzicht", "Kraamzorg de Waarden", and "Liemerscare").

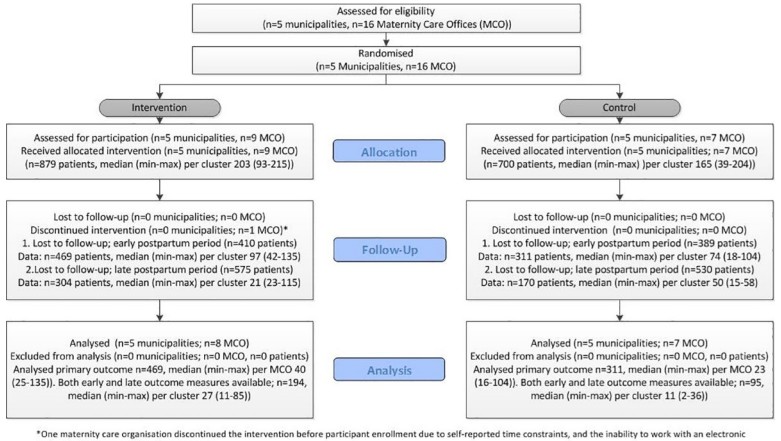

**Fig 1. CONSORT diagram showing flow of municipalities, maternity care organisations, and participants.**

## Trial design

The study design and rationale are briefly outlined below; a detailed study protocol was previously published (S1 File) [25]. This two-arm, parallel, pragmatic C-RCT was conducted with 12 independent maternity care organisations, which were geographically spread over the Netherlands with 16 offices (Fig 2).

Six out of ten municipalities within the HP4All-2 programme participated in this trial. Each municipality was divided into an intervention and a control cluster, to minimise the influence of geographical variation on the outcome measures. To avoid contamination between maternity care assistants (MCAs) working within the same maternity care organisation we allocated the intervention at the level of the maternity care organisation(s) per municipality rather than at the individual MCA or participant level. As such, a cluster could consist of multiple maternity care organisations.

## Participants

All pregnant women cared for by participating maternity care organisations with a scheduled home visit during pregnancy were eligible to take part in the study. Besides the unwillingness to sign a written informed consent form, there were no exclusion criteria for participants. Eligible participants were informed about the study and written informed consent was obtained prior to inclusion [25]. Participants were enrolled between July 2015 and December 2016. Follow-up data was collected until July 2017.

## Intervention and control conditions

The study included two practice changes: 1) implementation of a systematic risk assessment during pregnancy to identify medical, social and non-medical risk factors associated with adverse maternal health (control and intervention arm); and 2) provision of patient-centred care throughout pregnancy and the post-partum period by MCAs, as informed by their risk profile (intervention arm only) [25]. Participants were followed up until they were 12 weeks postpartum.

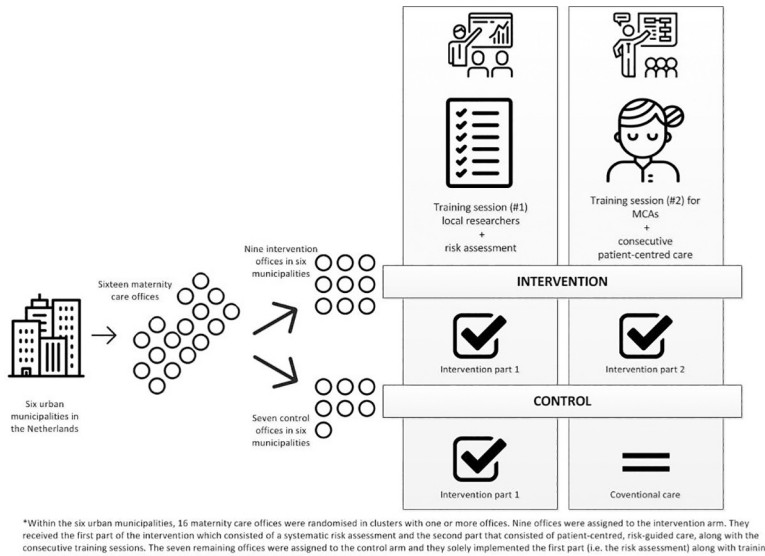

*Within the six urban municipalities, 16 maternity care offices were randomised in clusters with one or more offices. Nine offices were assigned to the intervention arm. They received the first part of the intervention which consisted of a systematic risk assessment and the second part that consisted of patient-centred, risk-guided care, along with the consecutive training sessions. The seven remaining offices were assigned to the control arm and they solely implemented the first part (i.e. the risk assessment) along with training session number one. Care provision in the control arm was based on conventional care.

**Fig 2. C-RCT design with intervention details.**

The systematic risk assessment was included in the standard home intake that is scheduled by maternity care organisations in the third trimester of pregnancy. The assessment was an electronic questionnaire and covered domains related to socioeconomic and psychological factors. The questionnaire consisted of the following components: 1) the Mind2Care (M2C) tool (a validated third trimester risk screening tool frequently used to assess psychological and psychosocial problems (www.mind2care.nl/home)); 2) the Maternal Empowerment Questionnaire (MEQ; a validated postpartum tool based on the World Health Organization (WHO) Responsiveness model used to assess the performance of maternity care based on clients' experiences [28]) with slight adjustments; and 3) additional questions regarding socioeconomic status, including questions regarding a person's financial situation, whether they have health care insurance, and their use of health care in the year prior to pregnancy Minor adjustments were made to ensure that the MEQ, which was originally designed to assess empowerment in the postpartum period, could be applied during pregnancy: the domains 'the future' and 'maternity care' were omitted, and minor grammatical adjustments were made in the remaining items as appropriate [25].

To properly carry out the risk assessment, professionals were trained in two separate sessions, each three hours long, referred to as 'Training Session 1a and 1b' (Fig 2). All trained professionals were MCAs, skilled nurses who have a lower secondary education degree for postpartum care. The first and core training was held prior to the start of participant enrolment and focused on effective use of the risk assessment tool and on building communication skills (Training 1a). The second training, after the first month of participant enrolment, was aimed at refreshing the obtained knowledge and providing a platform for discussing possible implementation issues that were encountered during patient enrolment (Training 1b). An established national educational organisation ("Kersten & van de Pol") assisted the research team in providing these training sessions [25]. Maternity care professionals who completed both Training Session number 1a and number 1b, and who enrolled participants in the study, are henceforth referred to as 'local researchers'.

For the second part (i.e. the actual intervention under study), the provision of patient-centred risk-tailored maternity care during the postpartum period, training sessions were held for all MCAs in the intervention arm prior to the start of the study (Training Session number 2). This training session covered knowledge and skills enabling MCAs to provide advice for additional care, to tailor maternity care provision to the individual needs and the existing resources of a participant, and to strengthen participants' own capabilities. Another established national educational organisation, ("Voorlichters Gezondheid"), assisted the research team in delivering Training Session 2. In addition to this accredited training session, templates were introduced to guide an MCA into taking appropriate action once a risk factor was identified. As such, each template served as a care pathway by directing the MCAs to, for example, a specific health care provider (i.e. midwife, obstetrician, or general practitioner), a public health care organisation, or an office for legal or financial support.

As an example, the template for addressing the risk factor 'irredeemable financial debts' guided the local researcher into taking the following actions during pregnancy: creating awareness among involved health care professionals (e.g. community midwife, general practitioner, MCAs) and applying for local funds that supply free postpartum packages consisting of, among other things, nappies and sanitary pads. In the control arm, professionals were blinded for the outcome of the risk assessment and they provided conventional care during pregnancy and the postpartum period (Fig 2).

## Outcomes

We hypothesised that our intervention would reduce the proportion of women with a low empowerment score postpartum. As such, the primary outcome was a low maternal

empowerment score in the early postpartum period, between one to two weeks after childbirth, as assessed by the validated MEQ [29]. A low empowerment score was defined as a MEQ-score beneath the 20th centile (rounded at one decimal) of empowerment scores in the control arm. Secondary outcome measures pertain to maternal health-related quality of life, maternal perceived health, maternal psychological health, maternal and neonatal health care utilisation postpartum, maternal smoking behaviour, and alcohol consumption during the late postpartum period, up and until twelve weeks after childbirth (S1 Table).

MCAs in the intervention arm filled out a questionnaire during the participant's early postpartum period. This questionnaire assessed the MCAs' understanding of the participant's obtained risk profile during pregnancy, and the relevance of the detected risk-profile for the care that they provided during the postpartum period.

All MCAs filled out the Measurement Instrument for Determinants of Innovations (MIDI); an instrument that measures determinants of innovations [30] at the end of the study period to assess the delivery of the intervention.

## Randomisation, implementation, and blinding

S1 Fig summarises the chronology of events, blinding, and any differences between the two arms [31]. The timeline cluster diagram is complementary to the CONSORT-flowchart for C-RCTs (Fig 1) [27]. At the start of the study, maternity care organisations were identified and recruited. Clusters were formed and consisted of one or more distinct maternity care organisations, depending on the expected number of eligible women within each organisation. Randomisation was then performed at the cluster level. A statistician who was not involved in the implementation of the trial and who was blinded to the identity of the clusters performed the randomisation procedure. After randomisation of the clusters, all eligible women received study information at first contact with a maternity care organisation and prior to the scheduled home visit. Hence, participants were blinded for allocation because they were unaware of the randomised design of the study, and as such of the alternative strategy. Blinding of MCAs and local researchers was not feasible due to the training related to our intervention. Therefore, recruitment of participants, the following baseline assessment, execution of the experimental intervention during pregnancy and the postpartum period, as well as outcome assessments in the postpartum period were partially blinded; the participant was unaware of the randomised design but the MCA was aware of the outcome of allocation.

## Sample size

An initial sample size of 1711 participants was calculated using an alpha of 0.05, a power of 80%, and assuming a lost to follow-up of 33% [25]. To adjust for clustering, an Intraclass Correlation Coefficient (ICC) of 0.05 was assumed in this sample size calculation, as is common in studies involving primary care practices [32, 33].

## Statistical methods

**Impact of the intervention.**   Descriptive data were compared to identify major differences between the two arms in baseline characteristics, pregnancy outcomes, and in uptake of maternity care.

Mixed-effects models were used to adjust for clustering within the (clusters of) maternity care organisations. Analyses were conducted with random effect for clusters only ('unadjusted models'), and additionally adjusted for selected covariates ('adjusted models'). Selection of covariates was based on a stepwise backward elimination using the Wald test and keeping a p value of 0.20 as a threshold for elimination. 'Parity' (multiparous vs primiparous), and 'single

parenthood' (living without a partner) were fixed prior to the stepwise modelling as these variables are incorporated in the national protocol for determining the amount of postpartum care provision and as such may affect maternal empowerment postpartum [22]. Multilevel mixed-effects logistic regression was used for binary outcomes, multilevel mixed-effects linear regression was used for continuous outcomes, and multilevel mixed-effects ordered regression was used for categorical outcomes. The main analyses were undertaken on an intention-to-treat (ITT) basis. The observed intraclass correlation coefficient was calculated using the adjusted model for the primary outcome.

*Subgroup analysis*. To assess whether there was a differential effect of the intervention on maternal empowerment according to whether risk factors had been identified during pregnancy, a subgroup analysis on participants with at least one identified factor was performed.

*Post-hoc analysis*. A higher-than-anticipated attrition rate was observed during the trial. In an attempt to address this we applied multiple imputation using chained equations, creating 20 unique datasets, to account for loss to follow-up and missing information [34]. Both predictor and outcome variables were included to inform the multiple imputation process and results across the sets were combined using Rubin's Rules [35]. A post-hoc sensitivity analysis using the multiple imputed data was performed to check whether this attrition rate affected the effect estimates of the primary outcome.

For all analyses, nominal significance level was 5%; with no adjustment for multiple testing. Analyses were performed using Stata version 15.

**Determinants of the delivery of the intervention.**   To evaluate the delivery of the intervention in the postpartum period, MCAs were asked to indicate if they were aware that a risk factor had been identified during pregnancy, and if they agreed that this detected risk had indeed been present throughout the postpartum period. To evaluate the determinants of implementation all local researchers were asked to fill out the MIDI questionnaire. The response scale consisted of a five point scale; 'totally disagree', 'disagree', 'neither agree nor disagree', 'agree', and 'totally agree'. All outcomes regarding these determinants of implementation are reported as frequencies and percentages.

## Results

### Cluster and participant flow

Sixteen maternity care offices from 12 independent maternity care organisations were recruited. In the intervention group, one maternity care office discontinued the intervention before participant enrolment due to difficulties with using electronic questionnaires (Fig 1). A total of 76 professionals from maternity care offices in both arms were recruited and trained extensively completing both Training Session 1a and 1b. All local researchers participated throughout the study period. Additionally, in the intervention arm, 385 MCAs were trained in Session number 2 and participated in the provision of patient-centred, risk-based maternity care during pregnancy and the postpartum period.

Together, the 15 maternity care offices that participated throughout the study period included 1579 participants: 879 participants in the intervention arm, and 700 participants in the control arm. Enrolment of participants stopped after December 2016 according to protocol [25].

Follow-up data from the early postpartum period, between one to two weeks after childbirth, was available for 469 participants (53%) in the intervention arm and 311 (44%) in the control arm (follow-up for both arms; 49%, attrition 51%). For all participants with follow-up data from the early postpartum period, data from the consecutive late postpartum period, six through twelve weeks after childbirth, was available for 304 participants (64%) in the

intervention arm and 170 participants (54%) in the control arm (overall follow-up for both arms; 30%, attrition 70%) (Fig 1).

Participants who were lost to follow-up during the early and late follow-up period more often: were first or second generation immigrants (early: 40% versus 24%; late: 37% versus 21%), had a low disposable household income (early 17% versus 7%; late 15% versus 6%), and were without a paid job during pregnancy (early follow-up 26% versus 17%; late follow-up 26% versus 12%) (S2 Table).

### Participant characteristics

Baseline characteristics assessed during pregnancy were similar for participants in the intervention and the control arm (Table 1); this similarity was also observed between clusters (S3 Table). The percentage of participants who had one or more risk factor(s) identified during pregnancy was similar between both arms: 40% and 39% in the intervention and the control arm, respectively (S4 Table). The most frequent risk factors were: fear for upcoming delivery (intervention 37%; control 34%); a self-reported concern about being prepared for care provision for the baby (intervention 27%; control 27%); and inadequate health literacy (intervention 17%; control 20%). Data regarding pregnancy outcomes and uptake of maternity care were available for 1445 participants (92%). There were no relevant differences in pregnancy outcomes or in the provided amount of maternity care between the intervention arm and the control arm (Table 2).

### Impact of the intervention

A low empowerment score in the early postpartum period was observed in 19.2% and 25.4% of all participants in the intervention and control arm, respectively. The intervention resulted in a significant reduction in the odds of having a low empowerment score [unadjusted OR 0.69 ((95% CI 0.48;0.99), P 0.047); adjusted OR 0.69 ((95% CI 0.47; 0.99), P 0.046)] (Table 3).

Analyses of predefined secondary outcomes showed no significant effects of the intervention on maternal health-related quality of life, perceived health, psychological health, maternal- and neonatal health care utilisation postpartum, and maternal smoking behaviour and alcohol consumption during the postpartum period (Table 3).

**Subgroup analysis.** The impact of the intervention on the incidence of a low maternal empowerment score was mainly attributable to its impact among participants with one or multiple identified risks during pregnancy (n = 412): aOR 0.61 ((95% CI 0.38; 0.98), P 0.043). There was no demonstrable effect of the intervention in the subgroup of women without identified risks during pregnancy (n = 368): aOR 0.78 ((95% CI 0.43; 1.41), P 0.407).

**Post-hoc sensitivity analysis.** Following imputation of missing data, the incidence of a low empowerment score was 20% and 23% in the intervention and control group, respectively. The estimated proportions of all predefined outcomes were more alike between the two arms in the imputed data than in the observed data (Table 3).

Within the post-hoc analysis using imputed data, the impact of the intervention was smaller than in the complete dataset and no longer statistically significant [OR 0.83 ((95% CI 0.61; 1.12); P 0.228); aOR 0.82 (95% CI 0.60; 1.10); P 0.182)].

### Determinants of the delivery of the intervention

**Delivery of the intervention during the postpartum period.** Of all participants in the intervention arm for whom follow-up data were available, 90% (n = 420) had a response from the MCA who provided maternity care to them. Of these participants, 55% (n = 233) had one or multiple identified risk factor(s) during pregnancy. The response from the MCAs indicated that they were aware of these risks in only 30% (n = 70) of all care provisions. Among those

**Table 1. Individual-level baseline characteristics assessed during pregnancy.**

| | Intervention (n = 879) | | Control (n = 700) | | |
|---|---|---|---|---|---|
| | **Mean** | **SD** | **Mean** | **SD** | **p-value** |
| Maternal age* | 31.5 | 4.8 | 31.7 | 4.4 | |
| | **N** | **%** | **N** | **%** | |
| Parity | | | | | |
| Primiparous | 446 | 50.7% | 360 | 51.4% | 0.786 |
| Multiparous | 433 | 49.3% | 340 | 48.6% | |
| Parenthood | | | | | |
| Single | 20 | 2.3% | 31 | 4.4% | 0.016 |
| Living together | 859 | 97.7% | 669 | 95.6% | |
| Immigrant status | | | | | |
| Non-immigrant | 595 | 68.7% | 468 | 67.1% | 0.057 |
| First generation | 135 | 15.6% | 97 | 13.9% | |
| Second generation | 136 | 15.7% | 132 | 18.9% | |
| Missing | 13 | 1.5% | 3 | 0.4% | |
| Health insurance | | | | | |
| No | 1 | 0.1% | 1 | 0.1% | 0.872 |
| Yes | 878 | 99.9% | 699 | 99.9% | |
| Education | | | | | |
| Lower | 50 | 5.7% | 43 | 6.1% | 0.925 |
| Intermediate | 538 | 61.2% | 428 | 61.1% | |
| High | 291 | 33.1% | 229 | 32.7% | |
| Net household income (euro/month) | | | | | |
| <1500 | 95 | 10.8% | 95 | 13.6% | 0.245 |
| 1500–3000 | 338 | 38.5% | 260 | 37.1% | |
| >3000 | 446 | 50.7% | 345 | 49.3% | |
| Paid job during pregnancy | | | | | |
| No | 201 | 22.9% | 145 | 20.7% | 0.304 |
| Yes | 678 | 77.1% | 555 | 79.3% | |
| Neighbourhood deprivation | | | | | |
| No | 517 | 58.8% | 370 | 52.9% | 0.018 |
| Yes | 362 | 41.2% | 330 | 47.1% | |
| Smoking during pregnancy | | | | | |
| No | 762 | 86.7% | 602 | 86.0% | 0.692 |
| Yes | 117 | 13.3% | 98 | 14.0% | |
| Alcohol use during pregnancy | | | | | |
| No | 693 | 78.8% | 565 | 80.7% | 0.358 |
| Yes | 186 | 21.2% | 135 | 19.3% | |
| Drug use during pregnancy | | | | | |
| No | 867 | 98.6% | 689 | 98.4% | 0.734 |
| Yes | 12 | 1.4% | 11 | 1.6% | |
| Risk factors detected | | | | | |
| No | 527 | 60.0% | 424 | 60.6% | 0.645 |
| Yes | 352 | 40.0% | 276 | 39.4% | |
| Indicated hours of maternity care | | | | | |
| 24–49 hours | 691 | 82% | 571 | 84% | 0.569 |
| >49 hours | 153 | 18% | 106 | 16% | |

(*Continued*)

**Table 1.** (Continued)

| | Intervention (n = 879) | | Control (n = 700) | | |
|---|---|---|---|---|---|
| | Mean | SD | Mean | SD | p-value |
| Missing | 34 | 4% | 23 | 3% | |

Mean* with SD or number with % (presented as percentage of non-missing values). Missing value percentage of total.

aware, 79% (n = 55) agreed that this risk factor had influenced the content of their care. This low awareness of MCAs was mostly due to lack of communication between MCAs responsible for care during pregnancy and those responsible for the postpartum period, as reported in the MIDI instrument.

**Determinants of implementation of the intervention.** At the end of the study period, 52 out of 76 trained local researchers (68%) filled out the MIDI questionnaire assessing the determinants of implementation of the intervention. Local researchers reported an overall positive feeling regarding the intervention. Sixty-seven percent of all local researchers agreed that the procedural clarity of the intervention was good, 56% agreed that the intervention was based on factually correct knowledge (i.e. the correctness of the intervention itself), and 57% reported a good compatibility of the intervention with their values and standard working patterns. When asked if they believed that the intervention was relevant for their client, only 8% disagreed. However, the evaluated determinants that were associated with themselves, as the adapting person, were scored less positively. Twenty-one percent of all local researchers reported little to no personal gain from the intervention with regard to improving their standard working patterns. Only 30% agreed with the following statement: "with this innovation I achieved risk-based, patient-centred care during pregnancy and the postpartum period".

**Table 2. Pregnancy outcomes and maternity care uptake postpartum.**

| | Intervention (n = 812) | | Control (n = 633) | | |
|---|---|---|---|---|---|
| | N | % | N | % | p-value |
| Gender newborn | | | | | |
| Male | 423 | 52.1% | 303 | 47.9% | 0.111 |
| Female | 389 | 47.9% | 330 | 52.1% | |
| Caesarean section | | | | | |
| No | 704 | 86.7% | 541 | 85.5% | 0.501 |
| Yes | 108 | 13.3% | 92 | 14.5% | |
| Preterm delivery | | | | | |
| No | 783 | 96.4% | 609 | 96.2% | 0.825 |
| Yes | 29 | 3.6% | 24 | 3.8% | |
| Low birth weight (<2500 grams) | | | | | |
| No | 773 | 96.0% | 601 | 95.7% | 0.757 |
| Yes | 32 | 4.0% | 27 | 4.3% | |
| Missing | 7 | 0.9% | 5 | 0.8% | |
| Provided hours of postpartum care | | | | | |
| <24 hours | 69 | 8.7% | 43 | 6.9% | 0.223 |
| 24–49 hours | 476 | 59.8% | 391 | 62.4% | |
| >49 hours | 251 | 31.5% | 193 | 30.8% | |
| Missing | 16 | 2.0% | 6 | 0.9% | |

Number with % (presented as percentage of non-missing values). Missing value percentage of total.

**Table 3. Impact of the intervention on primary and secondary outcomes.**

| Primary and secondary outcomes | | | | | |
|---|---|---|---|---|---|
| | | Proportions | | OR (95% CI[1]) | aOR (95% CI) |
| | N | Intervention | Control | | |
| Complete cases | | | | | |
| **Early postpartum period** | | | | | |
| *Primary outcome* | | | | | |
| Low empowerment score | 711 | | | | |
| Yes | | 19 | 25 | 0.69 (0.48; 0.99)* | 0.69 (0.47; 0.99)* |
| *Secondary outcomes* | | | | | |
| Maternal health related quality of life, cont. | 763 | | | | |
| mean (SE[2]) | | 0.82 (0.13) | 0.83 (0.13) | -0.01 (-0.03; 0.14) | -0.01 (-0.04; 0.11) |
| Maternal perceived health, cont. | 775 | | | | |
| mean (SE) | | 76.80 (13.1) | 76.43 (12.0) | 0.35 (-2.06; 2.76) | -0.07 (-1.99; 1.84) |
| **Late postpartum period** | | | | | |
| *Secondary outcomes* | | | | | |
| Maternal psychological health | 474 | | | | |
| EPDS[3]>12 | | 5 | 5 | 0.90 (0.35; 2.30) | 0.85 (0.31; 2.38) |
| Maternal health care utilisation | 413 | | | | |
| One visit | | 21 | 34 | 0.63 (0.32; 1.22) | 0.68 (0.35; 1.32) |
| Multiple visits | | 10 | 5 | | |
| Hospital admission | | 3 | 3 | | |
| Neonatal health care utilisation | 428 | | | | |
| One visit | | 12 | 12 | 1.29 (0.89; 1.89) | 1.32 (0.89; 1.94) |
| Multiple visits | | 28 | 30 | | |
| Hospital admission | | 6 | 7 | | |
| Smoking late postpartum period | 447 | | | | |
| Yes | | 6 | 5 | 1.13 (0.49; 2.56) | 0.96 (0.40; 2.28) |
| Alcohol use late postpartum period | 300 | | | | |
| Yes | | 39 | 32 | 1.34 (0.91; 2.00) | 1.40 (0.94; 2.10) |
| *Subgroup analysis: at least one risk factor in assessment* | | | | | |
| Low empowerment score | 412 | | | | |
| Yes | | 21 | 30 | 0.64 (0.40; 1.01) | 0.61 (0.38; 0.98)* |
| *Post-hoc analysis: imputed dataset* | | | | | |
| Low empowerment score | 711 | | | | |
| Yes | | 20 | 23 | 0.83 (0.61; 1.12) | 0.82 (0.60; 1.10) |

Number of complete cases per outcome (N) with proportions (out of 100) of primary and secondary outcomes; presented by early and late postpartum period (column: First) and by primary and secondary outcomes (second). Results from the multilevel mixed-effects logistic regression and multilevel mixed-effects ordered logistic regression are presented as (unadjusted) OR with 95% CI (first), and adjusted OR (aOR) with 95% CI (second). Results from multilevel mixed-effects linear regression are marked as cont. (continuous) and presented as β with 95% CI. Adjustment for parity, parenthood, household income, and educational attainment. Significant results are marked by an *.

[1]95% CI; 95% Confidence Interval.

[2]SE; Standard Error.

[3]EPDS; Edinburgh Postnatal Depression Scale.

## Discussion

This pragmatic C-RCT indicates that a structured risk assessment, including social and other non-medical determinants of health, followed by patient-centred maternity care may help

reduce the incidence of low maternal empowerment during the postpartum period. This effect was particularly seen among high-risk women, as detected by the assessment during pregnancy. Diminishing low empowerment postpartum may reduce the presence of significant stress, depressive symptoms, and improve the quality of care provided to the newborn, hereby minimising early adversity and existing inequalities in postpartum care.

This trial has several strengths. It contains several positive key characteristics of pragmatic trials: 1) comparison of a clinically relevant alternative to current practice; 2) a diverse and representative population of study participants; 3) heterogeneous practice settings similar to those where the aspired preventive care is to be set out; and 4) collection of data on a broad range of health outcomes [36, 37]. Furthermore, with this large scale randomised trial we are the first to evaluate the effectiveness of risk-guided, patient-centred care in increasing maternal self-efficacy.

In addition, analysis of determinants of implementation showed that the different intervention components were acceptable by local researchers from participating maternity care organisations. The combination of the good acceptability from health care professionals and the possibility of reducing low maternal empowerment scores, supports the potential for introducing this approach. Given that low maternal empowerment is associated with an increased risk of developing adverse maternal and child health, this intervention has the potential to reduce persisting inequalities in maternal and child health outcomes.

When interpreting our findings, a number of limitations also need to be taken into account. First, there was a considerable and selective loss to follow-up that has introduced a bias in our data. The loss to follow-up rate was particularly high among the less advantaged participants (S2 Table), a phenomenon that is often observed in clinical trials [38–40]. To address the resulting amount of missing data, including missing outcomes, we used multiple imputation. Baseline characteristics that were related to the loss to follow-up were included to inform the imputation model. However, concerns about validity persist because multiple imputation does not work well when data is missing not at random, hence producing biased estimates [41]. This complicated interpretation of the contrasting findings of the analyses of the imputed versus the original data, as both could be biased.

Second, the calculated sample size of 1711 participants was not achieved, possibly reducing the accuracy of our results. Formulas for calculating appropriate sample size in C-RCTs traditionally include inflation with the Variance Inflation Factor (VIF). This factor is calculated by multiplying the number of participants per cluster with the ICC [32, 33]. For our initial sample size calculation we anticipated an ICC of 0.05 [25]. However, the observed ICC from our primary model was negligible (ICC = 3.21 e-15), and as such the correlation between clusters was much less than expected. Sample size calculations using the observed rather than the anticipated ICC, together with the higher than expected attrition rate would have resulted in a lower required sample size (i.e. a total of 784 participants).

Third, the observed fidelity of our intervention was low, which might have affected its effectiveness [42]. We had intended to have MCAs tailor their maternity care to the identified risk factors in pregnancy. However, in only 40% of all postpartum care provisions that involved participants in whom risk factors had been identified during pregnancy, the MCA reported to be aware of these risks. This was unexpected, because due to the extensive training prior to the start of the intervention, all MCAs were accustomed with the importance of risk assessment and centring their care to meet these risks and the participants' own resources and capabilities. A possible explanation for the low fidelity might be the perceived little personal gain for professionals with regard to improving their standard working patterns.

Fourth, although the reduction in low empowerment scores in the intervention arm was statistically significant, it may not reflect a clinically important difference. Overall, women had

a strong sense of self efficacy during the initial assessment, and the difference in median scores between the two groups was small. Although the validity and reliability of the MEQ have been reported before as good [29], the questionnaire has not been tested on its responsiveness to change and the size of what may be considered a clinically meaningful difference has not been established.

Research has shown that it is challenging to show benefit of universally provided home visits regarding the wellbeing of mothers and babies, as underlined by the review of Yonemoto and colleagues [43]. They concluded that interventions aiming to improve maternal health outcomes in the postnatal period might be effective if subpopulations with a higher risk would be targeted [43]. It is in this regard important to note that we have previously demonstrated that low-SES women are less likely to receive postpartum care in the Netherlands, at the same time showing that lack of maternity care is associated with increased subsequent health care costs [23]. Hence, we intended to address a specific subpopulation of high-risk women during pregnancy, to enhance empowerment in those at overall greater risk of adverse health outcomes. Although study participants had a relatively high educational level, a high net income, and were more often non-immigrants compared to the general Dutch population, still 40% had one or more risk factors amendable to tailored care identified. Furthermore, the effect of the intervention was greatest in this subgroup of women with one or multiple risk factors, possibly reducing their risk of postpartum depression and increasing the quality of care provided to the newborn.

Future research should consider alternative designs to analyse the effectiveness of complex interventions into routine maternity care. Quasi-experimental studies or stepped-wedge trials may be better suited to assess impact of novel care adaptations. Alternatively, future research should include focus groups and interviews with women to hear from their perspective the challenges they face and the supports they need. Letting them be part of the design of the interventions could help reduce attrition and improve outcomes.

With this study, we have provided evidence that it is feasible to amend routine maternity care by introducing patient-centred informed by identification of predefined medical and non-medical risk factors, and that this may improve maternal empowerment in the postpartum period. Additional work is needed to ensure effective implementation of the different intervention aspects and to assess its efficiency.

## Supporting information

**S1 Fig. The timeline cluster diagram.**
(DOCX)

**S1 Table. Outcome measures at participant level with definitions.**
(DOCX)

**S2 Table. Baseline characteristics for participants who were lost to-follow up by timing of follow-up (early and late postpartum period).**
(DOCX)

**S3 Table. Baseline characteristics, presented by cluster and by intervention allocation.**
(DOCX)

**S4 Table. Self-reported identified risk factors(s), by intervention allocation.**
(DOCX)

**S1 File. Protocol HP4all maternity care.**
(PDF)

**S2 File. CONSORT 2010 checklist of information to include when reporting a randomised trial**\*.
(DOC)

## Acknowledgments

We express our gratitude to all maternity care organisations and women that were involved in our study.

## AcknowledgmentsRegistration

Netherlands Trial Registry (NTR) 6311.

Protocol: Lagendijk J, Been JV, Ernst-Smelt HE, Bonsel GJ, Bertens LCM, Steegers EAP. Client-tailored maternity care to increase maternal empowerment: cluster randomized controlled trial protocol; the Healthy Pregnancy 4 All-2 program. BMC Pregnancy and Childbirth 2019;19(1):4. doi: 10.1186/s12884-018-2155-9. Included in S1 File.

## Author Contributions

**Conceptualization:** Jacqueline Lagendijk, Meertien K. Sijpkens, Hiske E. Ernst-Smelt, Jasper V. Been, Eric A. P. Steegers.

**Data curation:** Jacqueline Lagendijk, Jasper V. Been.

**Formal analysis:** Jacqueline Lagendijk, Meertien K. Sijpkens, Jasper V. Been.

**Funding acquisition:** Hiske E. Ernst-Smelt, Eric A. P. Steegers.

**Investigation:** Jacqueline Lagendijk.

**Methodology:** Jacqueline Lagendijk, Meertien K. Sijpkens, Jasper V. Been, Eric A. P. Steegers.

**Project administration:** Jacqueline Lagendijk, Hiske E. Ernst-Smelt.

**Resources:** Jacqueline Lagendijk.

**Supervision:** Hiske E. Ernst-Smelt, Sarah B. Verbiest, Jasper V. Been, Eric A. P. Steegers.

**Validation:** Jacqueline Lagendijk, Meertien K. Sijpkens, Hiske E. Ernst-Smelt, Sarah B. Verbiest, Jasper V. Been.

**Visualization:** Jacqueline Lagendijk, Meertien K. Sijpkens.

**Writing – original draft:** Jacqueline Lagendijk, Meertien K. Sijpkens, Jasper V. Been.

**Writing – review & editing:** Jacqueline Lagendijk, Meertien K. Sijpkens, Hiske E. Ernst-Smelt, Sarah B. Verbiest, Jasper V. Been, Eric A. P. Steegers.

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
