## [Decision Letter · Decision Letter 0]

16 Mar 2020

PONE-D-19-23605

Enhancing maternal empowerment postpartum: a cluster randomised controlled trial

PLOS ONE

Dear Miss Lagendijk,

Thank you for submitting your manuscript to PLOS ONE. After careful consideration, we feel that it has merit but does not fully meet PLOS ONE’s publication criteria as it currently stands. Therefore, we invite you to submit a revised version of the manuscript that addresses the points raised during the review process.

The manuscript has been evaluated by two reviewers; their comments are available below.

The reviewers find the study of relevance but have raised some comments regarding the need for additional clarifications on methodological aspects and further discussion of the limitations of the work and to place the intervention in context.

Could you please revise the manuscript to address the items raised?

We would appreciate receiving your revised manuscript by Apr 28 2020 11:59PM. Please include the following items when submitting your revised manuscript:

We look forward to receiving your revised manuscript.

Kind regards,

Iratxe Puebla

Deputy Editor-in-Chief, PLOS ONE

Journal Requirements:

2. Thank you for submitting your clinical trial to PLOS ONE and for providing the name of the registry and the registration number. The information in the registry entry suggests that your trial was registered after patient recruitment began. PLOS ONE strongly encourages authors to register all trials before recruiting the first participant in a study.

1) your reasons for your delay in registering this study (after enrolment of participants started);

2) confirmation that all related trials are registered by stating: “The authors confirm that all ongoing and related trials for this drug/intervention are registered”.

Please also ensure you report the date at which the ethics committee approved the study as well as the complete date range for patient recruitment and follow-up in the Methods section of your manuscript.

4. Your ethics statement must appear in the Methods section of your manuscript. If your ethics statement is written in any section besides the Methods, please move it to the Methods section and delete it from any other section. Please also ensure that your ethics statement is included in your manuscript, as the ethics section of your online submission will not be published alongside your manuscript.

Reviewers' comments:

Reviewer's Responses to Questions

**Comments to the Author**

1. Is the manuscript technically sound, and do the data support the conclusions?

Reviewer #1: Yes

Reviewer #2: Yes

2. Has the statistical analysis been performed appropriately and rigorously? 

Reviewer #1: Yes

Reviewer #2: Yes

3. Have the authors made all data underlying the findings in their manuscript fully available?

Reviewer #1: Yes

Reviewer #2: Yes

4. Is the manuscript presented in an intelligible fashion and written in standard English?

Reviewer #1: Yes

Reviewer #2: Yes

5. Review Comments to the Author

Reviewer #1: The statistical analysis, including the employment of mixed-effects models to adjust for clustering, covariates selection, and multiple imputation for missing data, were generally appropriate.

The limitation had generally been appropriately discussed and the conclusion was sufficiently conservative in view of these highlighted limitations.

I just have the following minor concerns:

Although the author has highlighted the lack of an established clinically meaningful difference for the MEQ (Line 337-339), a discussion of the possible interpretation and potential implication of a low empowerment score would be needed to inform the proper interpretation of the result by the readers.

The much less positive result from the perspective of the adapting persons (as highlight in Line 289-293), would need to be better discussed in the discussion and to be highlighted as a potential concern that needs to be recognised and addressed.

Reviewer #2: Thank you for this manuscript on maternal empowerment in the third trimester and in the first 12 weeks post-partum. I have some comments to make, whih I hope you will consider in your revision.

Line 62-66: Mention some specific mother and infant health outcomes related to self-efficacy, to help put the potential value of this intervention into context.

Line 97: Were there any exlusion criteria?

Line 104: Give some examples of the social and non-medical risk factors being identified.

Line 115: Provide more detail on what you mean by making "slight adjustments" to the questionnaire.

Line 116: You mention that professionals were trained in assessment. What was the background/profession of these professionals?

Line 226: If participants consented to follow-up at both time points, when they were approached in pregnancy, the attrition figures should also reflect this. In addition to the figures on follow-up between 2 and 12 weeks post-partum, include overall follow-up/attrition figures, i.e. 304/879 (34.6% follow-up, 65.4% attrition) and 170/700 (24.3% follow-up, 75.7% attrition). Were any reasons for dropping out of the study given by participants? Or were they simply not able to be contacted at 12 weeks post-partum?

Line 357:You mention that it is "feasible to" implement this intervention, but perhaps you could emphasise further that more work is needed before it would be advisable to implement this intervention.

Line 362: Put the abbreviations in alphabetical order.

Table 1: Can you insert p-values, to show the similarities/differences between groups?

Table 3: Can the proportions be given out of 100, rather than out of 1? Some abbreviations in the table need to be written out in full at the bottom, e.g. EPDS, SE, etc. Can significant results be indicated, to include p-values?

6. PLOS authors have the option to publish the peer review history of their article (what does this mean?). If published, this will include your full peer review and any attached files.

Reviewer #1: No

Reviewer #2: No

---

## [Author Response · Author response to Decision Letter 0]

16 Apr 2020

Jacqueline Lagendijk

Erasmus Medical Centre

P.O. Box 2040 3000 CA Rotterdam

The Netherlands

Dr J. Heber

Editor-in-Chief

PLOS ONE

April 16th, 2020

Dear Dr Heber, 

Enclosed please find our revised manuscript entitled “Enhancing maternal empowerment postpartum: a cluster randomised controlled trial” (PONE-D-19-23605). Thank you for this opportunity. 

We have very carefully considered the valuable feedback from the reviewers and have revised our manuscript accordingly. In the following point-by-point reply, we first reproduce the reviewer’s feedback in full before detailing our responses in bold, and the revised parts of our manuscript in a cursive bold font. Please note that the page and line numbers refer to the manuscript with tracked changes. 

Yours Sincerely,

On behalf of all authors,

Jacqueline Lagendijk, research fellow

 

Point by point reply;

Reviewer 1: 

The statistical analysis, including the employment of mixed-effects models to adjust for clustering, covariates selection, and multiple imputation for missing data, were generally appropriate.

The limitation had generally been appropriately discussed and the conclusion was sufficiently conservative in view of these highlighted limitations.

Response: we thank the reviewer for the kind words and the time taken to provide feedback on our manuscript. 

I just have the following minor concerns:

Although the author has highlighted the lack of an established clinically meaningful difference for the MEQ (Line 337-339), a discussion of the possible interpretation and potential implication of a low empowerment score would be needed to inform the proper interpretation of the result by the readers.

Response: in line with this recommendation, we have added information about the potential implications of (low) maternal empowerment to the first paragraph of the introduction. 

Introduction, page 4, line 63-67. 

“Maternal self-efficacy is a belief held by mothers of their capabilities to perform specific tasks in taking care of the newborn. This self-efficacy, or empowerment, is an important predictor of a successful transition to motherhood (4). Reduced self-efficacy postpartum, or a low maternal empowerment score, reduces the practice of exclusive breastfeeding, the quality of care provided to the newborn, and it increases the risk of maternal depression (4-6).” 

To further aid proper interpretation of the results, we have provided some examples in the fourth paragraph of the discussion. 

Discussion, page 16, line 332-337. 

“This pragmatic C-RCT indicates that a structured risk assessment, including social and other non-medical determinants of health, followed by patient-centred maternity care may help reduce the incidence of low maternal empowerment during the postpartum period. This effect was particularly seen among high-risk women, as detected by the assessment during pregnancy. Diminishing low empowerment postpartum may reduce the presence of significant stress, depressive symptoms, and improve the quality of care provided to the newborn, hereby minimising early adversity and existing inequalities in postpartum care.”

The much less positive result from the perspective of the adapting persons (as highlight in Line 289-293), would need to be better discussed in the discussion and to be highlighted as a potential concern that needs to be recognised and addressed.

Response: we agree with the reviewer that the less positive perspective of the professional is an important limitation of our trial. A possible explanation for the low fidelity might be the perceived little gain for professionals’ standard working scheme, as mentioned in the method section. We have highlighted the need to do further research to increase the feasibility and efficiency of our intervention in the last paragraph op the discussion. 

Discussion, page 17, line 367-374. 

“Third, the observed fidelity of our intervention was low, which may have affected its effectiveness. We had intended to have MCAs tailor their maternity care to the identified risk factors in pregnancy. However, in only 40% of all postpartum care provisions that involved participants in whom risk factors had been identified during pregnancy, the MCA reported to be aware of these risks. This was unexpected, because due to the extensive training prior to the start of the intervention, all MCAs were accustomed with the importance of risk assessment and centring their care to meet these risks and the participants’ own resources and capabilities. A possible explanation for the low fidelity might be the limited personal gain experienced by professionals from the intervention with regard to improving their standard working patterns.”

Discussion, page 18, line 399-402. 

“With this study, we have provided evidence that it is feasible to amend routine maternity care by introducing patient-centred care informed by identification of predefined medical and non-medical risk factors, and that this may improve maternal empowerment in the postpartum period. Additional work is needed to ensure effective implementation of the different intervention aspects and to assess its efficiency.”

Reviewer: 2

Thank you for this manuscript on maternal empowerment in the third trimester and in the first 12 weeks post-partum. I have some comments to make, whih I hope you will consider in your revision.

Response: we thank the reviewer for the time taken to provide feedback on our manuscript. 

Line 62-66: Mention some specific mother and infant health outcomes related to self-efficacy, to help put the potential value of this intervention into context. 

Response: we thank the reviewer for this valuable recommendation. We have elaborated more on the relation between maternal self-efficacy and health outcomes in the revised introduction. To further underline the possible adverse effects of low empowerment we have now note some related adverse outcomes to our revised discussion. 

Introduction, page 4, line 63-67. 

“Maternal self-efficacy is a belief held by mothers of their capabilities to perform specific tasks in taking care of the newborn. This self-efficacy, or empowerment, is an important predictor of a successful transition to motherhood (4). Reduced self-efficacy postpartum, or a low maternal empowerment score, reduces the practice of exclusive breastfeeding, the quality of care provided to the newborn, and it increases the risk of maternal depression (4-6).” 

Discussion, page 16, line 332-337. 

“This pragmatic C-RCT indicates that a structured risk assessment, including social and other non-medical determinants of health, followed by patient-centred maternity care may help reduce the incidence of low maternal empowerment during the postpartum period. This effect was particularly seen among high-risk women, as detected by the assessment during pregnancy. Diminishing low empowerment postpartum may reduce the presence of significant stress, depressive symptoms, and improve the quality of care provided to the newborn, hereby minimising early adversity and existing inequalities in postpartum care.”

Line 97: Were there any exlusion criteria?

Response: besides the unwillingness to sign a written informed consent form, there were no exclusion criteria for participants. There were exclusion criteria for participating maternity care organisations as listed in the study protocol. Maternity care organisations could not participate if they were not able to conduct a scheduled home visit during pregnancy for all participants. Full details about the different in- and exclusion criteria are listed in the published protocol. 

Methods, page 6, line 122-127

“All pregnant women cared for by participating maternity care organisations with a scheduled home visit during pregnancy were eligible to take part in the study. Besides the unwillingness to sign a written informed consent form, there were no exclusion criteria for participants. Eligible participants were informed about the study and written informed consent was obtained prior to inclusion (25). Participants were enrolled between July 2015 and December 2016. Follow-up data was collected until July 2017.”

Line 104: Give some examples of the social and non-medical risk factors being identified.

Response: in line with this recommendation we have added some examples of social and non-medical risk factors that were assessed during pregnancy. 

Methods, page 7, line 136-143. 

“The questionnaire consisted of the following components: 1) the Mind2Care (M2C) tool (a validated third trimester risk screening tool frequently used to assess psychological and psychosocial problems (www.mind2care.nl/home)); 2) the Maternal Empowerment Questionnaire (MEQ; a validated postpartum tool based on the World Health Organization (WHO) Responsiveness model used to assess the performance of maternity care based on clients' experiences) with slight adjustments; and 3) additional questions regarding socioeconomic status, including questions regarding a person’s financial situation, whether they have health care insurance, and their use of health care in the year prior to pregnancy.”

Line 115: Provide more detail on what you mean by making "slight adjustments" to the questionnaire.

Response: the validated Maternal Empowerment Questionnaire, measures a woman’s empowerment postpartum. The risk assessment in this trial was undertaken during pregnancy rather than during the postpartum period, which led to the necessity to make slight adjustments. Two out of five domains of the MEQ were omitted for this assessment after consulting the authors (i.e. domains “the future” and “maternity care”) as these were not applicable during pregnancy. All other questions were grammatically adjusted for application during pregnancy, rather than after childbirth. The details about these adjustments, as listed above, are mentioned in the protocol of our trial. 

Methods, page 7, line 134-146

“The systematic risk assessment was included in the standard home intake that is scheduled by maternity care organisations in the third trimester of pregnancy. The assessment was an electronic questionnaire and covered domains related to socioeconomic and psychological factors. The questionnaire consisted of the following components: 1) the Mind2Care (M2C) tool (a validated third trimester risk screening tool frequently used to assess psychological and psychosocial problems (www.mind2care.nl/home)); 2) the Maternal Empowerment Questionnaire (MEQ; a validated postpartum tool based on the World Health Organization (WHO) Responsiveness model used to assess the performance of maternity care based on clients' experiences (28)) with slight adjustments; and 3) additional questions regarding socioeconomic status, including questions regarding a person’s financial situation, whether they have health care insurance, and their use of health care in the year prior to pregnancy. Minor adjustments were made to ensure that the MEQ, which was originally designed to assess empowerment in the postpartum period, could be applied during pregnancy: the domains ‘the future’ and ‘maternity care’ were omitted, and minor grammatical adjustments were made in the remaining items as appropriate.”

Line 116: You mention that professionals were trained in assessment. What was the background/profession of these professionals?

Response: all trained professionals were Maternity Care Assistants (MCAs). MCAs are skilled nurses who have a lower secondary education degree for postpartum care. In the Netherlands, maternity care is provided by MCAs at home under supervision of the community midwife. The duration of maternity care is indexed in advance by an MCA for each woman individually during a scheduled home visit during the second or third trimester of pregnancy. Following delivery, a MCA visits and supports the family at home on a daily basis for the first eight to ten consecutive days. 

Methods, page 7, line 147-153

“To properly carry out the risk assessment, professionals were trained in two separate sessions, each three hours long, referred to as ‘Training Session 1a and 1b’ (Figure 1). All trained professionals were MCAs, skilled nurses who have a lower secondary education degree for postpartum care. The first and core training was held prior to the start of participant enrolment and focused on effective use of the risk assessment tool and on building communication skills (Training 1a). The second training, after the first month of participant enrolment, was aimed at refreshing the obtained knowledge and providing a platform for discussing possible implementation issues that were encountered during patient enrolment (Training 1b).”

Line 226: If participants consented to follow-up at both time points, when they were approached in pregnancy, the attrition figures should also reflect this. In addition to the figures on follow-up between 2 and 12 weeks post-partum, include overall follow-up/attrition figures, i.e. 304/879 (34.6% follow-up, 65.4% attrition) and 170/700 (24.3% follow-up, 75.7% attrition). Were any reasons for dropping out of the study given by participants? Or were they simply not able to be contacted at 12 weeks post-partum?

Response: consent for all questionnaires was obtained at a single time point; during the scheduled home visit in the second or third trimester in pregnancy. 

Unfortunately, there was a high loss to follow-up especially for the last questionnaire between six and 12 weeks postpartum. MCAs assisted participants with the first questionnaire, during the home visit. The second questionnaire was filled out during the early postpartum period when MCAs were still present on a daily basis. The third questionnaire was sent six weeks after delivery with two reminders, each after three weeks, to the home address of all participants. The lack of routine maternity care at this time point might have influenced the response from participants. 

For clarification we have added the attrition rate of the different time point to the result section. 

Results, page 11, line 257-262. 

“Follow-up data from the early postpartum period, between one to two weeks after childbirth, was available for 469 participants (53%) in the intervention arm and 311 (44%) in the control arm (follow-up for both arms; 49%, attrition 51%). For all participants with follow-up data from the early postpartum period, data from the consecutive late postpartum period, six through twelve weeks after childbirth, was available for 304 participants (64%) in the intervention arm and 170 participants (54%) in the control arm (overall follow-up for both arms; 30%, attrition 70%) (figure 2).”

Line 357: You mention that it is "feasible to" implement this intervention, but perhaps you could emphasise further that more work is needed before it would be advisable to implement this intervention.

Response: we agree with the reviewer that more research is needed before implementing this intervention. We have highlighted the possibility to amend routine practices with a risk assessment and following tailored care throughout pregnancy and the postpartum period. However, the efficiency of the different intervention aspects need to be clarified. We have revised the last paragraph of our discussion to better highlight the limitations and the need for further research.

Discussion, page 18, line 399-402. 

“With this study, we have provided evidence that it is feasible to amend routine maternity care by introducing patient-centred informed by identification of predefined medical and non-medical risk factors, and that this may improve maternal empowerment in the postpartum period. Additional work is needed to ensure effective implementation of the different intervention aspects and to assess its efficiency.

Line 362: Put the abbreviations in alphabetical order. 

Response: alphabetical ordering was applied to the list of abbreviations. 

Page 19, line 404-420

“ABBREVIATIONS

95% CI 95% Confidence Interval

aOR adjusted odds ratio 

C-RCT Cluster-randomised controlled trial

EPDS Edinburgh Postnatal Depression Scale

ICC Intraclass correlation coefficient

ITT Intention to treat

HP4All Healthy Pregnancy 4 All

MCA Maternity care assistant

MEQ Maternal empowerment questionnaire

MIDI Measurement Instrument for Determinants of Innovations

M2C Mind2Care

OR Odds ratio

SE Standard Error

SES Socioeconomic status

VIF Variance inflation factor

WHO World Health Organisation”

Table 1: Can you insert p-values, to show the similarities/differences between groups?

Response: thank you for this recommendation, we have revised table 1 and table 2 accordingly. 

Results, page 12, line 277

“Table 1: Individual-level baseline characteristics assessed during pregnancy.”

 Intervention (n=879) Control (n=700) 

 Mean SD Mean SD p-value

Maternal age* 31.5 4.8 31.7 4.4 

 N % N % 

Parity 

Primiparous 446 50.7% 360 51.4% 0.786

Multiparous 433 49.3% 340 48.6% 

Parenthood 

Single 20 2.3% 31 4.4% 0.016

Living together 859 97.7% 669 95.6% 

Immigrant status 

Non-immigrant 595 68.7% 468 67.1% 0.057

First generation 135 15.6% 97 13.9% 

Second generation 136 15.7% 132 18.9% 

Missing 13 1.5% 3 0.4% 

Health insurance 

No 1 0.1% 1 0.1% 0.872

Yes 878 99.9% 699 99.9% 

Education 

Lower 50 5.7% 43 6.1% 0.925

Intermediate 538 61.2% 428 61.1% 

High 291 33.1% 229 32.7% 

Net household income (euro/month) 

<1500 95 10.8% 95 13.6% 0.245

1500-3000 338 38.5% 260 37.1% 

>3000 446 50.7% 345 49.3% 

Paid job during pregnancy 

No 201 22.9% 145 20.7% 0.304

Yes 678 77.1% 555 79.3% 

Neighbourhood deprivation 

No 517 58.8% 370 52.9% 0.018

Yes 362 41.2% 330 47.1% 

Smoking during pregnancy 

No 762 86.7% 602 86.0% 0.692

Yes 117 13.3% 98 14.0% 

Alcohol use during pregnancy 

No 693 78.8% 565 80.7% 0.358

Yes 186 21.2% 135 19.3% 

Drug use during pregnancy 

No 867 98.6% 689 98.4% 0.734

Yes 12 1.4% 11 1.6% 

Risk factors detected 

No 527 60.0% 424 60.6% 0.645

Yes 352 40.0% 276 39.4% 

Indicated hours of maternity care 

24-49 hours 691 82% 571 84% 0.569

>49 hours 153 18% 106 16% 

Missing 34 4% 23 3% 

Results, page 13, line 280. 

“Table 2: Pregnancy outcomes and maternity care uptake postpartum.”

 Intervention (n=812) Control (n=633) 

 N % N % p-value

Gender newborn 

Male 423 52.1% 303 47.9% 0.111

Female 389 47.9% 330 52.1% 

Caesarean section 

No 704 86.7% 541 85.5% 0.501

Yes 108 13.3% 92 14.5% 

Preterm delivery 

No 783 96.4% 609 96.2% 0.825

Yes 29 3.6% 24 3.8% 

Low birth weight (<2500 grams) 

No 773 96.0% 601 95.7% 0.757

Yes 32 4.0% 27 4.3% 

Missing 7 0.9% 5 0.8% 

Provided hours of postpartum care 

<24 hours 69 8.7% 43 6.9% 0.223 

24-49 hours 476 59.8% 391 62.4% 

>49 hours 251 31.5% 193 30.8% 

Missing 16 2.0% 6 0.9% 

Table 3: Can the proportions be given out of 100, rather than out of 1? Some abbreviations in the table need to be written out in full at the bottom, e.g. EPDS, SE, etc. Can significant results be indicated, to include p-values?

Response: in line with the recommendation of the reviewer we have given proportions out of 100, listed the abbreviations in the footnote and placed an * after all significant results. 

Results, page 14, line 305. 

“Table 3: Impact of the intervention on primary and secondary outcomes.”

Primary and secondary outcomes

 Proportions OR (95% CI1) aOR (95% CI)

 N Intervention Control 

Complete cases 

Early postpartum period 

Primary outcome 

Low empowerment score 711 

Yes 19 25 0.69 (0.48; 0.99)* 0.69 (0.47; 0.99)*

Secondary outcomes 

Maternal health related quality of life, cont. 763 

mean (SE2) 0.82 (0.13) 0.83 (0.13) -0.01 (-0.03; 0.14) -0.01 (-0.04; 0.11)

Maternal perceived health, cont. 775 

mean (SE) 76.80 (13.1) 76.43 (12.0) 0.35 (-2.06; 2.76) -0.07 (-1.99; 1.84)

Late postpartum period 

Secondary outcomes 

Maternal psychological health 474 

EPDS3>12 5 5 0.90 (0.35; 2.30) 0.85 (0.31; 2.38)

Maternal health care utilisation 413 

One visit 21 34 0.63 (0.32; 1.22) 0.68 (0.35; 1.32)

Multiple visits 10 5 

Hospital admission 3 3 

Neonatal health care utilisation 428 

One visit 12 12 1.29 (0.89; 1.89) 1.32 (0.89; 1.94)

Multiple visits 28 30 

Hospital admission 6 7 

Smoking late postpartum period 447 

Yes 6 5 1.13 (0.49; 2.56) 0.96 (0.40; 2.28)

Alcohol use late postpartum period 300 

Yes 39 32 1.34 (0.91; 2.00) 1.40 (0.94; 2.10)

Subgroup analysis: at least one risk factor in assessment

Low empowerment score 412 

Yes 21 30 0.64 (0.40; 1.01) 0.61 (0.38; 0.98)*

Post-hoc analysis: imputed dataset

Low empowerment score 711 

Yes 20 23 0.83 (0.61; 1.12) 0.82 (0.60; 1.10)

---

## [Decision Letter · Decision Letter 1]

29 Oct 2020

Risk-guided maternity care to enhance maternal empowerment postpartum: a cluster randomized controlled trial

PONE-D-19-23605R1

Dear Dr. Lagendijk,

We’re pleased to inform you that your manuscript has been judged scientifically suitable for publication and will be formally accepted for publication once it meets all outstanding technical requirements.

Kind regards,

Anna Palatnik, M.D.

Academic Editor

PLOS ONE

Additional Editor Comments (optional):

Thank you for addressing reviewers' comments.

We are happy to accept your paper for publication.

Reviewers' comments:

Reviewer's Responses to Questions

**Comments to the Author**

1. If the authors have adequately addressed your comments raised in a previous round of review and you feel that this manuscript is now acceptable for publication, you may indicate that here to bypass the “Comments to the Author” section, enter your conflict of interest statement in the “Confidential to Editor” section, and submit your "Accept" recommendation.

Reviewer #2: All comments have been addressed

2. Is the manuscript technically sound, and do the data support the conclusions?

Reviewer #2: Yes

3. Has the statistical analysis been performed appropriately and rigorously? 

Reviewer #2: Yes

4. Have the authors made all data underlying the findings in their manuscript fully available?

Reviewer #2: Yes

5. Is the manuscript presented in an intelligible fashion and written in standard English?

Reviewer #2: Yes

6. Review Comments to the Author

Reviewer #2: All comments have been satisfactorily addressed , with thanks. I have nothing further to add. With kind regards.

7. PLOS authors have the option to publish the peer review history of their article (what does this mean?). If published, this will include your full peer review and any attached files.

Reviewer #2: No

---

## [Editor Report · Acceptance letter]

9 Nov 2020

PONE-D-19-23605R1 

Risk-guided maternity care to enhance maternal empowerment postpartum: a cluster randomized controlled trial 

Dear Dr. Lagendijk:

I'm pleased to inform you that your manuscript has been deemed suitable for publication in PLOS ONE. Congratulations! Your manuscript is now with our production department. 

Kind regards, 

on behalf of

Dr. Anna Palatnik 

Academic Editor

PLOS ONE